# Dual Role of Chitin as the Double Edged Sword in Controlling the NLRP3 Inflammasome Driven Gastrointestinal and Gynaecological Tumours

**DOI:** 10.3390/md20070452

**Published:** 2022-07-11

**Authors:** Chetan Roger Dhanjal, Rathnamegha Lingamsetty, Anooshka Pareddy, Se-Kwon Kim, Ritu Raval

**Affiliations:** 1Department of Biotechnology, Manipal Institute of Technology (MIT), Manipal Academy of Higher Education (MAHE), Manipal 576104, Karnataka, India; chetan.dhanjal@gmail.com (C.R.D.); rathnamegha@gmail.com (R.L.); anooshka.pareddy@gmail.com (A.P.); 2Manipal Biomachines, Manipal Institute of Technology (MIT), Manipal Academy of Higher Education (MAHE), Manipal 576104, Karnataka, India; 3Department of Marine Science and Convergence Engineering, College of Science and Technology, Hanyang University, Erica55 Hanyangdae-ro, Sangnok-gu, Ansan-si 11558, Gyeonggi-do, Korea

**Keywords:** NLRP3, chitin derivatives, chitosan, chito-oligosaccharides, gastric cancers, gynaecological cancers

## Abstract

The role of NLRP3 in the tumour microenvironment is elusive. In some cancers, the activation of NLRP3 causes a worse prognosis and in some cancers, NLRP3 increases chances of survivability. However, in many cases where NLRP3 has a protumorigenic role, inhibition of NLRP3 would be a crucial step in therapy. Consequently, activation of NLRP3 would be of essence when inflammation is required. Although many ways of inhibiting and activating NLRP3 in cancers have been discussed before, not a lot of focus has been given to chitin and chitosan in this context. The availability of these marine compounds and their versatility in dealing with inflammation needs to be investigated further in relation with cancers, along with other natural extracts. In this review, the effects of NLRP3 on gastrointestinal and gynaecological cancers and the impact of different natural extracts on NLRP3s with special emphasis on chitin and chitosan is discussed. A research gap in using chitin derivatives as anti/pro-inflammatory agents in cancer treatment has been highlighted.

## 1. Introduction

The NLRP3 inflammasome is an essential component of the innate immunity system. This inflammasome consists of three major proteins, the adaptor apoptosis-associated speck-like protein (ASC, which contains a caspase activation and recruitment domain, CARD), pro-caspase 1 (the effector molecule) and NLRP3 (containing pyrin domain 3) [1]. The pyrin domain of the NLRP3 interacts with ASC to start the inflammasome assembly. The mechanism responsible for activation is shown in Figure 1 [2]. Studies investigated the positive and negative effects that NLRP3 and its components have on specific types of cancers and gastrointestinal and gynaecological cancers seem to have a more pronounced effect [3]. 

Current options for cancers treatment such as radiation therapy and chemotherapy, due to their non-selectivity and potency of the drugs can lead to detrimental long-term effects in patients. Even with breakthrough natural drugs such as paclitaxel, there are significant side effects such as hepatotoxicity, neurotoxicity and cardiotoxicity [4]. However, studies on other natural extracts such as curcumin show positive results as an anticancer drug. 

The abundance of natural extracts from plants, fungi and marine ecosystems provides an optimistic number of extracts that can be studied for their anti-cancer properties. Marine ecosystems in particular, with more than 80% of the oceans being undiscovered, can be a valuable resource for anti-cancer extracts. For example, chitin, an abundant polysaccharide in nature, mostly occurs in the exoskeletons of crustaceans [5]. 

The possibility of using chitin and its derivatives as a potential anti-cancer drug due to its effects on the NLRP3 inflammasome and its components is underexplored. The authors suggest that due to the varying effects on the components of the inflammasome, chitin and its derivatives can be used as a potential anti-cancer drug or as an adjuvant. This paper further discusses the parts of an NLRP3 inflammasome that affect different kinds of cancer and how chitin and its derivatives influence the NLRP3 inflammasome. 

Pro-inflammatory responses have been shown to have a relationship with the proliferation of cancer cells, and cancer growth in general [3]. These pro-inflammatory responses can cause complications if not dealt with early on, leading to chronic inflammation and complications with the cancer. Pro-inflammatory factors such as NF-κB are shown to be constitutively expressed in tumour cells and targeting similar pro-inflammatory factors can be an approach to curb cancer proliferation [6].

Due to the activation of the inflammasome, pyroptosis (inflammasome mediated cell death) of macrophages, dendritic cells and sometimes other types of cells can take place [7]. As discussed previously, the effect of inflammation can affect the proliferation and growth of cancers. There have been studies showing that inflammation is helpful for a good prognosis in some cancers and that anti-inflammation is helpful for a good prognosis in other cancers [8,9]. Thus, the activation and inactivation of the inflammasome and its components can impact the cancer positively or negatively depending on the type of cancer. In this review we discuss the effects of NLRP3 on gastrointestinal and gynaecological cancers.

## 2. NLRP3 Inflammasome and Its Importance in Gastrointestinal and Gynaecological Cancers

### 2.1. Gastrointestinal Cancers

Gastrointestinal cancers refer to the cancers that occur along the gastrointestinal tract. Gastrointestinal cancers are caused either by rare hereditary forms or the more common sporadic forms. These causes are further exacerbated by lifestyle choices such as smoking and diet, infection by virus or bacteria or other variables like age [10]. 

According to the GLOBOCAN database, in 2018, there were 18.1 million new cancer diagnoses and 9.6 million cancer deaths globally. Gastrointestinal (GI) tract malignancies were responsible for more than 15% of cancer incidence (Figure 2) and 17% of cancer-related mortality [11]. The role of NLRP3 in gastrointestinal cancers seems to generally be pro-tumoral with exceptions in colorectal cancer where there are cases of anti-tumoral activity as well.

#### 2.1.1. Colorectal Cancer (CRC)

The development of colorectal cancer begins with the conversion of normal epithelial cells to adenomas which undergo further alterations to become carcinomas which can further develop into metastatic cancer, which is affected by chromosomal and microsatellite instability and the CpG island methylator phenotype [12]. This cancer was the third highest in number of cases, and fourth in mortality with 1.9 million cases and 0.9 million deaths in 2020 [13]. The role of NLRP3 in colorectal cancer is controversial and seems to be a double-edged sword. When colitis was induced by dextran sulphate sodium (DSS), NLRP3^−/−^ and IL18^−/−^ mice were more susceptible to tumorigenesis [14]. Conversely, a study conducted in China showed that patients with low NLRP3 production had a better prognosis of stage I or II CRC [15]. A similar study by Wang et al. “the association of aberrant expression of NLRP3 and p-S6K1 in colorectal cancer” on CRC tissue samples from a total of 135 patients, showed the same results [16]. In another study, it was concluded that NLRP3 upregulation was directly correlated with epithelial to mesenchymal transition, a process that plays a key role in tumour progression [17]. Other research conducted in 2019 on C57 BL/6 mice that were injected with azoxymethane intraperitoneally, showed that inhibiting NLRP3 protected against tumorigenesis in colitis-associated CRC [18]. However, the actual role of NLRP3 in the progression of CRC remains elusive as there are studies that prove the protagonist or antagonist role it plays.

#### 2.1.2. Pancreatic Cancer 

Pancreatic ductal adenocarcinoma (PDA) was reported to be the 14th most common type of cancer with 495,773 cases worldwide, but with an extremely high mortality rate, with 466,003 deaths [19]. The hypothesised causes of PDA include the activation of RAS oncogene, activation of Notch genes which strongly upregulate NF-κB, and the stimulation of COX-2 which helps in prostaglandin formation [20]. NLRP3 helps with tumour progression and negatively impacts a patient’s prognosis due to the production of IL-1. A study conducted on NLRP3^−/−^ mice showed that inhibiting NLRP3 by deletion or by pharmacological inhibition, resulted in a better survivability of the mice. NLRP3 signalling in macrophages drives the change of the adaptive immune response to be tolerogenic as it promotes the differentiation of existing T-cells into tumour-promoting T-cells [21]. Similar to CRC, NLRP3 is also involved in the epithelial to mesenchymal transition (EMT) as shown by a study where the inhibition of NLRP3 resulted in reversion of EMT, inhibition of cell invasion and proliferation [22]. A study carried out by Boone et al. in 2019 on mouse models showed that platelet NLRP3 activation caused aggregation and increased tumour growth in murine models [23]. In a study by Yew et al. in 2020, the results on the pancreatic cell lines Panc10.05, PANC1 and SW1990, when treated with the NLRP3 inhibitor MCC950, showed notable reduction in cell viability [12]. Another study by Lui et al. in 2020 on MNS (3,4-methylenedioxy-β-nitrostyrene) as an inhibitor of NLRP3, showed that the downregulation of NLRP3 showed an inhibitory effect on the metastasis and proliferation of PDAC cells [24,25].

#### 2.1.3. Gastric Cancer

Gastric cancer (GC) is a common malignancy in the digestive system. Most studies related to GC and NLRP3 are related to incidence of gastritis and GC due to *Heliobacter pylori* infection. Due to the PAMPs released by *H*. *pylori*, the NLRP3 inflammasome pathway gets activated and produces IL-1β and IL-18. These cytokines are significantly linked with GC. As the infection has found many ways to evade the immune system, the IL-1β levels persist which leads to gastric atrophy, the initial step in carcinogenesis [26]. Activation of NLRP3 happens through the induction of intracellular ROS production [27], potassium efflux and lysosomal destabilisation [26]. Contradictorily, a study showed that *H. pylori* avoids activation of NLRP3 [28]. However, the role of NLRP3 in GC is clearly to promote tumorigenesis in most cases and targeting NLRP3 maybe a good treatment option. 

#### 2.1.4. Hepatic Cancer

Hepatic cancer or hepatocellular carcinoma had 905,677 new cases which made up 4.7% of all new cancer cases in 2020, and 830,180 deaths accounting for about 8.3% of deaths caused by cancer globally in 2020 [11]. This makes it a cancer with a very high mortality rate. The effect of NLRP3 is not well defined in hepatic cancer, with it showing a positive effect and negative effect on the same. 

A study conducted investigating the effect of NLRP3 on hepatic cancer showed that treatment of HepG2 cell line with 17β-estradiol which upregulates NLRP3, shows phagocytosis of these cells and downregulation of protective autophagy, both of which combined leads to suppression of proliferation of hepatic cancer [29]. This was further supported by another study showing that NLRP3 activity was either severely downregulated or completely lost in late-stage hepatic cancer, and its low activity is correlated with poor pathogen differentiation [30].

On the contrary, NLRP3 has also been shown to be downregulated when administered anti-tumoral miR-223 to prevent proliferation of hep3B cells [31], suppressed by celastrol and metformin which are known to downregulate markers of angiogenesis, proliferation and metastasis [31], and has been shown to be present in high levels when administering diethylnitrosamine, a potent carcinogen [32]. 

### 2.2. Gynaecological Cancers

Gynaecological cancers are cancers that originate in the female reproductive system, this includes endometrial, cervical, ovarian, vulvar, vaginal and fallopian tube associated [33]. According to the GLOBACAN database, in women, the incidence of gynaecological cancers is 14.4% (Figure 2) and the mortality is more than 13% [11]. These cancers have vague symptoms that can be easily dismissed and are often diagnosed late. Therefore, further research is required to understand the incidence of these cancers and establish methods of prevention and cure. 

#### 2.2.1. Endometrial Cancer

Endometrial cancer is the sixth most prevalent cancer in women and the 15th most common cancer in the general population, with 417,367 new cases and 97,370 deaths in 2020 [11]. Many studies have been conducted to attribute the tumorigenesis of endometrial cancers to NLRP3. However here, tumour progression is also largely associated with oestrogen. In a study conducted with 31 patients, oestrogen increased the growth of endometrial cancer cells by upregulating NLRP3 expression via the oestrogen receptor (ERβ) [34]. Studies on NLRP3 also show the relation between Erβ and the inflammasome, simultaneously correlating this to tumour activity [34]. The levels of oestrogen and therefore NLRP3 activation seem to affect the proliferation. 

Another study discussing a contrary result show that although NLRP3 is present in higher quantities in endometrial cancer cells, caspase-1 and IL-1β are not expressed in a correspondingly high fashion. Hydrogen treatment of these cell lines showed an increase in mitochondrial ROS which facilitated NLRP3-dependent pyroptosis of the cancer cells and showing a protective effect on normal epithelial endometrial cells [35].

#### 2.2.2. Ovarian Cancer

Ovarian cancer was the third most common gynaecological cancer, with 313,959 cases and 207,252 deaths in 2020 worldwide [11]. The onset of ovarian cancer usually happens after menopause, but cases have occurred with early menarche, and ages before 55. The cause of ovarian cancer is not well known, but there are relations between mutated BRCA1 and BRCA2 genes (genes which help in repair of DNA) and occurrence of ovarian cancer [36]. Only a few studies have been conducted that elucidate the relationship between NLRP3 and ovarian cancer prognosis, but they all point towards a higher level of NLRP3- and NLRP3-associated components being related to a worse prognosis [1,37]. There is a need for further research into the downregulation of NLRP3 and its effects on ovarian cancer.

#### 2.2.3. Cervical Cancer

Cervical cancer was the second most common gynaecological cancer, with 604,127 cases and 341,831 deaths in 2020. Cervical cancer is highly correlated with the Human Papilloma Virus (HPV) with around 99% of all cases being linked to persistent infection via HPV [11]. The onset of cervical cancer usually happens between the ages of 35 and 44 [38]. In a study by Pontillo et al. in 2016, a notable association was found between the *rs*107454558 variant of NLRP3 and HR-HPV. The study showed that HPV/host interactions are affected by inflammasome genetics in terms of cancer progression, virus persistence and susceptibility [39]. 

Another study by He et al. showed that CD200, a membrane glycoprotein (from the immunoglobulin superfamily) and CD200Fc, the soluble formulation fusion protein demonstrated suppression in inflammatory activity in the NLRP3 and TLR 4-NF-κB inflammation pathways in CaSi and SiHa cell lines [40]. However, there is a need for further research into the downregulation of NLRP3 and its effects on cancer. 

## 3. Effects of Cancer Treatment

### 3.1. Chemical Treatments 

Although the type of treatment prescribed by a medical professional depends on the type and stage of the tumour, chemotherapy and radiation therapy are most often the first recommended course of treatment for most cancers. Additionally, chemotherapy and radiotherapy doses and aggressiveness of the treatment also depends on the same factors [41]. Apart from the side effects felt by a patient during cancer therapy namely nausea, vomiting, hair loss, decreases immunity and fatigue [42], the damage done to the body exhibits long term side effects, or “late effects’’, such as nerve damage, cognitive difficulties, hearing problems, increased risk to other cancers etc., [43,44,45]. In most cases these late effects are expected to stop but can remain for months or years after cancer therapy but also may cause permanent damage to the patients reproductive, circulatory, respiratory, and sensory systems [42]. In a study by Keikhaei et al., late effects of cancer treatment among children exhibited as hypothyroidism, scoliosis, reduced growth rate and sensorineural hearing loss among others [46].

### 3.2. Natural Extracts for Treatment

Despite success of the natural chemotherapeutic drug paclitaxel, its non-selective cytotoxicity remains to be a problem [47]. Other natural extracts show promise and potential in cancer treatment without significant detrimental effects. Their selective cytotoxicity towards cancer cells can be exploited as a cancer treatment alternative. In a study to investigate the therapeutic potential of a novel method using curcumin extracted from *Curcum longa*’s rhizome, no significant cytotoxicity was observed in non-cancer NIH3T3 cells but displayed a significant cytotoxic effect when used on HeLa cell line. The study also showed that this novel method displayed a large decrease in expression of NF-κB suggesting anti-inflammatory effects [48]. 

Similarly, studies on various natural extracts yielded analysis of a multitude of potential anti-cancer properties of extracts from plants [49], cyanobacteria [50]. Natural extracts showing both anti-inflammatory and anti-cancer properties can be an effective option for treatment of gastrointestinal and gynaecological cancers. A few natural extracts and their anti-cancer and anti-inflammatory response are shown in Table 1. 

The abundance of marine sources for mentioned natural extracts can prove to be a key point to note. These natural sources provide for a great variation and diversity in structure and function as therapeutic drugs [57]. From marine sources, chitin (β-(1,4)-poly-N-acetyl-D-glucosamine) is an abundant polysaccharide found in the shells or exoskeletons of crustaceans [5]. Chitooligosaccharide (COS) is a derivative of chitin which has proven to show anti-cancer properties. COS additionally shows the downregulation of transcriptional and translational expression levels of TNF-α, IL-6, iNOS and COX-2, suggesting anti-inflammatory effects. The abundance of chitin from marine sources, anti-inflammatory and selective anti-cancer effects can be a powerful combination for a possible successful cancer treatment [55]. 

## 4. The Role of Chitin and Its Derivatives 

Natural extracts can affect cancer via multiple mechanisms, which include targeting specific points of inflammasome activation. Chitin and its derivatives such as chitosan and n-acetylated chitosan are similarly able to induce a response to the NLRP3 inflammasome in a multitude of ways [58,59,60]. These responses can be both pro-inflammatory and anti-inflammatory. We could potentially harness the effect of these molecules to influence the response in a way that is beneficial to specific types of cancer. 

The role of NLRP3 in cancers is dual and it has hostile and amicable effects toward different cancers [61]. Therefore, we can leverage the different effects chitin and different derivatives of chitin have, to assist in treatment of cancer on a case-by-case basis.

### 4.1. Effect of Chitin

Chitin is a very widely available polymer in nature, second to only cellulose. It is found mainly in fungal cell wall and exoskeleton of crustaceans. It is a long chain polymer of N-acetylglucosamine with β-(1–4) linkages [62]. Although chitin is not found in mammals, there is a presence of chitinases in mammals [63]. Chitin can also prove to be an avenue to recognise fungal pathogens due to lack of chitin in mammals normally. Studies conducted on human cancer cell lines have shown chitin to inhibit cell growth. Cytotoxicity was also shown against the Hep2 line at a concentration of 2000 μg/mL [64].

Chitin has been shown to have pro-inflammatory and anti-inflammatory effects when introduced, which is based on the preparation and size of molecule [65]. A study performed on C57BL/6 mice (TLR2 or TLR4 null mice and control mice) showed the effects of different sizes of chitin on the inflammatory response and categorised chitin as small chitin (SC) and intermediate chitin (IC) [66]. 

Chitin molecules of size 40–70 µm are IC and chitin molecules of size lesser than 40 µm are SC. Chitin molecules larger than 70 µm are classified as big chitin (BC) and was found to be inert to the inflammatory response [66]. Due to the size of BC, recognition and activity of any enzymes was ineffective so they are inert but if they get degraded by mammalian chitinase, that can generate IC and SC. In the same study, IC induced a pro-inflammatory response via TLR2 and NF-κB-dependent pathways for TNF production [66]. 

SC was able to induce an anti-inflammatory response via the production of IL-10 which is an anti-inflammatory cytokine [66]. IL-10 can suppress the immune response via binding to the heterodimeric receptors IL-10R1 and IL-10R2, which when activated triggers the JAK/STAT signalling pathway that serves to inhibit phagocytosis, presenting receptors and inhibition of release of pro-inflammatory cytokines [67,68] (Figure 3).

Chitin is mostly just recognised in mammalian cells due to the absence of chitin normally. The LYSMD3 domain is a pattern recognition receptor that is capable of binding chitin and sensing it. Other pattern recognition receptors that have affinity to chitin and derivatives of chitin are FIBCD1, NKR-P1, RegIIIγ and galectin-3 [69,70,71,72,73].

Chitin can occur in three types of crystal structures, α-, β- and γ-forms. The α-form is the most abundant in nature. However, due to the arrangement of these crystals, there is more inter-molecular bonding, albeit these bonds stabilise the structure, they hinder the ability of chitin to dissolve in many common solvents [74]. This prompts the usage of chitosan instead of chitin for pharmaceutical use as chitosan can dissolve in more solvents depending on the degree of deacetylation [75]. 

### 4.2. Effect of Chitosan

Chitosan is a polymer of chitin that is formed when chitin is deacetylated, yielding β-1,4-D-glucosamine [76]. It can be synthesised via biological and chemical methods, with chemical methods being employed more often in the industry due to its increased bioactivity and easier processing [77]. The chemical methods of production via by-products of crustaceans involve a three-step process that includes demineralisation, deproteination and deacetylation to finally yield chitosan [76].

The purity and molecular weight of chitosan strongly influences the immune response that it induces [78,79] and having a high degree of purity while using chitin or its derivatives is a must.

The molecular weight of chitosan is also a major factor in affecting the immune response caused. In a study conducted, it was found that chitosan with molecular weight greater than 29.2 kDa inhibited inflammation by downregulating the MAPK signalling pathway, inhibiting NF-κB activation and prevents expression of iNOS which brings down NO levels, and stopping the production of TNF-α and IL-6 [79]. These factors combined reduce the activation of the NLRP3 inflammasome (Figure 4).

On the contrary, chitosan of molecular weight lesser than 29.2 kDa acts in a pro-inflammatory manner by increasing phosphorylation of JNK of the MAPK pathway, enhancing NF-κB activation and iNOS expression causing an increase in the levels of NO, and increasing the production of TNF-α and IL-6 by binding to CR3, TLR4 and CD14 on macrophages. All these factors combined increase the activation of the NLRP3 inflammasome [79]. In a study on Hep2 and RD cell lines, chitosan showed cytotoxicity against these cell lines, and had a better IC than chitin and irradiated chitin [64].

Studies on chitosan often only characterise MW or DDA in high detail and leave out the other and fail to fully characterise the features of chitosan. This can lead to ambiguity in results. Studies have found that DDA does not make a difference in the immunoreactive properties of chitosan [76]. It is also suggested that endotoxin presence can interfere with the properties of chitosan. Another factor to consider is that as the DDA of chitosan approaches 100% its solubility increases that can play a role in being more bioavailable and affecting its properties during delivery [80].

Further treatment of the chitosan polymer can lead to the formation of chitosan nanoparticles and chitosan scaffolds. Although the role of DDA in the effect of chitosan polymer affecting the immune system has contradictory and ambiguous results, there are studies investigating the properties of the scaffolds and the nanoparticle showing a correlation between DDA and the properties of the material.

In a study showing the effect of DDA in chitosan nanoparticles significantly changes the viability of RAW 264.7 cells, with lower DDA showing higher cytotoxicity [81,82]. Interestingly, the physicochemical properties of the nanoparticles themselves seemed to have a larger effect on the properties of the nanoparticles.

In chitosan scaffolds, it was seen that the effect of DDA influences the type of macrophage recruited to the site of implant, consequently affecting whether there is a pro or anti-inflammatory effect [83].

### 4.3. Effect of Chitooligosaccharides

Chitooligosaccharides (COS) are degraded products of chitosan by enzymatic or chemical methods. Chitosan, like its precursor is still insoluble and not easily absorbed into the body. However, COS is soluble and easily absorbed by the gastrointestinal tract, after which it affects the target cells. This implies that COS can be utilised for a myriad of applications by retaining the bioactive properties of chitosan [84].

As explored previously, the antitumoral effect of COS is attributed to recruitment of T-cells and immunoregulation. We propose that alongside this, there is an inflammation/inflammasome-dependent manner by which COS and variations of COS are able to affect cancer [5]. It is likely that a combination of the two of these factors leads to the potent anti-cancer properties that these molecules possess.

This hypothesis can be further supported by studies conducted on the types of cancers described previously. A study conducted by Shen et al. showed that the use of COS with 23.99 kDa molecular weight, markedly reduced the proliferation of HepG2 cells by inhibiting the NF-κB [85,86]. Studies were conducted on colorectal cancer cells showing that utilisation of COS between 1 and 3 kDa reduced pro-inflammatory cytokine production by the inhibition of MM2 and iNOS, which led to a preventative effect [84].

A study using the LPS induced inflammation model showed that N-acetyl-chitooligosaccharides (NACOS) were able to attenuate inflammation via NLRP3 by suppressing NF-κB by preventing degradation of IκB, preventing phosphorylation of MAPK which is important for signalling inflammation-related genes, and also promotes the degradation and prevents activation of the NLRP3 inflammasome by suppressing production of pro IL-1β and by inhibiting caspase-1 cleavage into the active P20 subunit [87].

## 5. Conclusions

As discussed in the paper, cancer has an intricate tie with inflammation and components of the inflammasome. A change in the inflammatory response can aggravate or mitigate the proliferation of cancer cells. Thus, the NLRP3 inflammasome and its components can be targeted to reduce the effect of cancers.

Chitin and its derivatives are able to upregulate and downregulate the effect of the NLRP3 inflammasome based on its preparation, and these different reactions can be utilised to successfully target a broad range of cancers. Out of chitin, chitosan and COS, COS seems to be the best approach to actual products due its solubility being the highest, enabling it to be delivered more efficiently when compared to the other two. These specific preparations can be used on a case-by-case basis to help mitigate the negative effects of cancer and can potentially be used as a treatment or an adjuvant to cancer treatment.

Studies also suggest the possible usage of chitin and its derivatives as therapeutics to diseases that have inflammation-driven or inflammation-related complications. In cancers where inflammation can greatly impact the effect of survival, having a solution to modulate inflammation to a more suitable level could potentially change the efficacy of treatment.

## Figures and Tables

**Figure 1 marinedrugs-20-00452-f001:**
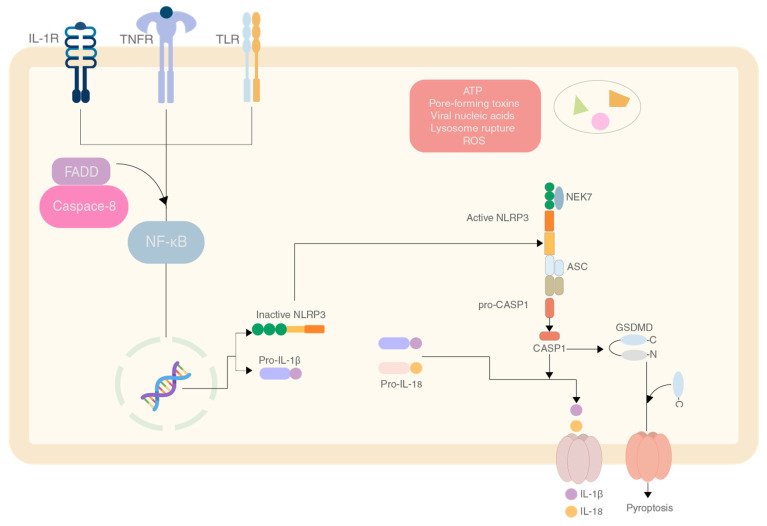
The mechanism of production and activation of the NLRP3 inflammasome by various effector molecules.

**Figure 2 marinedrugs-20-00452-f002:**
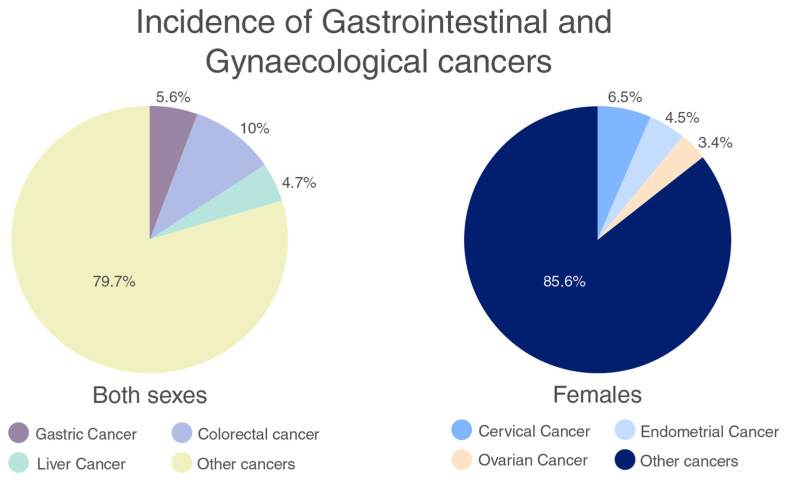
Incidence of gastric and gynaecological cancers [11].

**Figure 3 marinedrugs-20-00452-f003:**
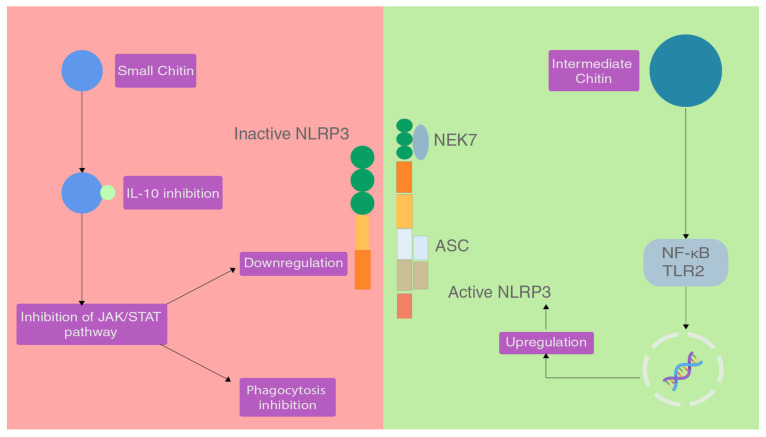
The effect of different sizes of chitin on the NLRP3 inflammasome.

**Figure 4 marinedrugs-20-00452-f004:**
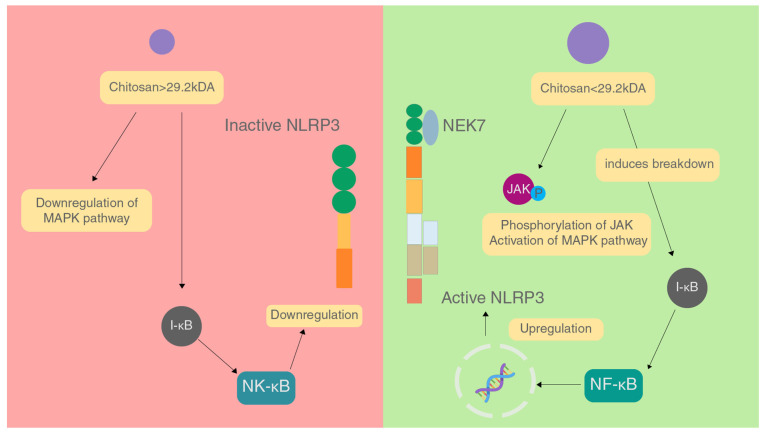
Effect of different sizes of chitosan on the NLRP3 inflammasome.

**Table 1 marinedrugs-20-00452-t001:** Natural extracts and their anti-cancer efficacies.

S.No	Extracts	Source	Effects on Inflammation	Effect on Cancer	Type of Cancer	Reference
1	Sesquiterpenoids	Soft coral (*Sinularia leptoclados)*	Inhibition and up-regulation of pro-inflammatory iNOS and cyclooxygenase-2 (COX-2) proteins	Not cytotoxic	HeLa	[51]
2	Diterpenoids	Dongsha Atoll soft coral (*Sinularia flexibilis)*	Pro-inflammatory iNOS and COX-2 (cyclooxygenase-2) proteins are inhibited and up-regulated	Moderate cytotoxicity	HeLa, SK-Hep1, and B16 cancer cells	[52]
3	Phenolic compound oleocanthal	Virgin olive oil	Reduced levels of cytokines, LTs, CRP, and PGs, as well as reduction of iNOS, COX, 5-LOX and NFB activity	Induces apoptosis	CRC cell line	[53]
4	Curcumin	*Curcuma longa*	Blocking of NF-κB activation	Selective cytotoxic effect	HeLa	[48,54]
5	Chito-oligosaccharides	Crustaceans, insects and fungi	Translational and transcriptional expression levels of iNOS, COX-2, TNF- and IL-6 are reduced.	Anticancer effect exhibited	HeLa and SW480 cell lines	[55]
6	Taxol	Pacific yew tree	Activates NLRP3 in macrophages causing inflammation	Strong anticancer effect	Breast, ovarian, pancreatic cancer	[56]

## Data Availability

Not applicable.

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
