# Peer review of "Dual Role of Chitin as the Double Edged Sword in Controlling the NLRP3 Inflammasome Driven Gastrointestinal and Gynaecological Tumours"

_marinedrugs, 2022, doi:10.3390/md20070452_

Round 1

Reviewer 1 Report

The manuscript discussed the recent progress in the effects of NLRP3 on gastrointestinal and gynaecological cancers and the impact of different natural extracts on NLRP3 with special emphasis on chitin and chitosan. The research gap in using chitin derivatives as anti/pro-inflammatory agents in cancer treatment has been high-lighted. Overall, the manuscript is clearly expressed. The presentation and structure is also acceptable. Most importantly, this review provided future perspectives in depth about the specific synthesis of Chitin and chitosan, which will help mitigate the negative effects of cancer treatment. So I suggest this manuscript to be published on marine drugs after minor revision.

1)        There are some small mistakes in the article, such as the inconsistent format of the references. Authors are strongly suggested to check it carefully.

2)        An overall schematic illustration about the idea of this review is highly suggested.

Author Response

Reply to Reviewer 1

The manuscript discussed the recent progress in the effects of NLRP3 on gastrointestinal and gynaecological cancers and the impact of different natural extracts on NLRP3 with special emphasis on chitin and chitosan. The research gap in using chitin derivatives as anti/pro-inflammatory agents in cancer treatment has been high-lighted. Overall, the manuscript is clearly expressed. The presentation and structure is also acceptable. Most importantly, this review provided future perspectives in depth about the specific synthesis of Chitin and chitosan, which will help mitigate the negative effects of cancer treatment. So I suggest this manuscript to be published on marine drugs after minor revision.

  • There are some small mistakes in the article, such as the inconsistent format of the references. Authors are strongly suggested to check it carefully.

Reply: As suggested the format has been revised and the changes have been highlighted.

  • An overall schematic illustration about the idea of this review is highly suggested.

Reply: As suggested the illustration has been added.

Reviewer 2 Report

The manuscript has tried to discuss the rule of chitin and its derivatives in NLRP3-mediated tumors. According to the text, it seems that it is NLRP3 itself that shows such behavior, while chitin and its derivatives have no direct effect on the tumors. Therefore I do not feel that the title of the manuscript has been chosen properly. 

By the way, most of the text deals with not related information to the title. For example up to page 7, one can not find any correlation between the rule of chitin and cancer. 

Author Response

Reply to Reviewer 2

The manuscript has tried to discuss the rule of chitin and its derivatives in NLRP3-mediated tumors. According to the text, it seems that it is NLRP3 itself that shows such behavior, while chitin and its derivatives have no direct effect on the tumors. Therefore I do not feel that the title of the manuscript has been chosen properly. 

Reply: As suggested the title has been modified to “Dual role of Chitin as the Double Edged Sword in Controlling the NLRP3 Inflammasome Driven Gastrointestinal and Gynaecological Tumors” where the cancers and the role of NLRP3 inflammasome is introduced initially followed by the studies conducted in regulating the NLRP3 inflammassome.

By the way, most of the text deals with not related information to the title. For example up to page 7, one can not find any correlation between the rule of chitin and cancer. 

Reply: As suggested the title has been modified. We have cited the studies  conducted on the cell lines with chitin and its derivatives after introducing various aspects of cancers and the role of NLRP3 inflammasome.

Reviewer 3 Report

Manuscript Revision

Title: Dual role of Chitin and Its Derivatives in NLRP3 Mediated Tumours.

Personally, I find the text interesting. The authors have carried out an interesting work on evaluating the ability of chitin and its derivates to prevent tumors. However, the manuscript presents a series of drawbacks that need to be corrected before the manuscript can be published. In the following lines, I will explain the main mistakes found.

Line 15. “evidently”. Avoid using this type of vocabulary. It may sound very pretentious.

Line 41. A reference is placed in the text after citing this figure. Is it an adapted figure?

Line 44. Rewrite. Some sentences of the text are worded in a complex way.

The introduction is a bit short. It should be enlarged.

Line 65. Missing white space before reference. Check that this mistake is not repeated again along with the text (e.g., line 81).

Line 71. Unify the references.

Line 73 – 74. These types of sentences are more suitable in the introduction. Assess unifying both sections.

Figure 2. Where do these data come from? In addition, it is necessary to improve the quality of the figure. What is the percentage of each color? Is it global or they are data from a region?

Line 82. The information in the manuscript is not well arranged. This paragraph is a clear example. This information would be more appropriate clarify before figure 2.

Line 93. “2020 [13]. The” instead of “2020 [13] The”.

Line 96. Delete the year.

Line 101. “Another study” is repeated in very close sentences. Replace with a synonym so that the text does not sound repetitive.

Line 124. “()”?? Is there information missing?

Line 128. Abbreviations only need to be clarified the first time they appear in the text.

Line 161. “here”. Rewrite.

Line 174. “worldwide [33].” Instead of “worldwide[33].”.

In section 2, there is no mention of chitin or its derivatives. These compounds are supposed to be the topic of the manuscript according to the title and the introduction. This mistake is repeated throughout the manuscript, so the title might be not appropriate. Otherwise, it is necessary to focus the information on this type of compounds.

Line 200 – 203. References missing.

Line 204 – 205. Rewrite.

Line 207. “[41]. In” instead of “[41].In”.

Section 3.1. What does this section have to do with the dual role of chitin and its derivatives in nlrp3-mediated tumors? This is a mistake common to almost all the manuscripts. Table 1 has also the same problem.

Line 230 – 239. This paragraph is almost the only one that justifies the title of the manuscript in sections 2 and 3.

Line 243. [55-57] instead of [55]–[57].

Line 254 – 255. Reference?

Line 261. “was”? Check verb tense.

Line 266-267. There cannot be paragraphs of only two lines.

Section 4.1. The information is disorganized and not well focused.

Line 339. “Utilized by us”. Rewrite. The same type of mistake in line 341 (“we can use…”).

Line 345. “complications. In” instead of “complications). In”

FINAL REMARKS

In my opinion, the authors have carried out a really interesting study, with promising expectations for future research. The manuscript is clear and well written. However, there are some issues that should be improved. Mainly, much of the text focuses on the mechanism of action of tumors, but not on the involvement of chitin and its derivates, which was the objective/theme of the manuscript according to the title. Therefore, I am suggesting MAJOR REVISIONS and RECONSIDERATION. The study should be improved before publication.

Author Response

Reply to Reviewer 3 

Line 15. “evidently”. Avoid using this type of vocabulary. It may sound very pretentious.

Reply: As suggested “evidently “ has been deleted. 

Line 41. A reference is placed in the text after citing this figure. Is it an adapted figure?

Reply:

 Line 44. Rewrite. Some sentences of the text are worded in a complex way.

Reply: The figure is our artwork , the reference as suggested has been deleted.

The introduction is a bit short. It should be enlarged.

Reply: As suggested the introduction is elaborated.

Line 65. Missing white space before reference. Check that this mistake is not repeated again along with the text (e.g., line 81).

Reply: As suggested the references have been rectified.

Line 71. Unify the references.

Reply: As suggested the references have been unified.

Line 73 – 74. These types of sentences are more suitable in the introduction. Assess unifying both sections.

Reply: As suggested the modification has been made.

Figure 2. Where do these data come from? In addition, it is necessary to improve the quality of the figure. What is the percentage of each color? Is it global or they are data from a region?

Reply: As suggested , the percentage have been included and the incidence is of global observation which is included in the text.

Line 82. The information in the manuscript is not well arranged. This paragraph is a clear example. This information would be more appropriate clarify before figure 2.

Reply: As suggested the modification has been made.

 Line 93. “2020 [13]. The” instead of “2020 [13] The”.

Reply: As suggested the modification has been made.

Line 96. Delete the year.

Reply: As suggested the modification has been made.

Line 101. “Another study” is repeated in very close sentences. Replace with a synonym so that the text does not sound repetitive.

Reply: As suggested the modification has been made.

Line 124. “()”?? Is there information missing?

Reply: As suggested the modification has been made.

Line 128. Abbreviations only need to be clarified the first time they appear in the text.

Reply: As suggested the modification has been made.

Line 161. “here”. Rewrite.

Reply: As suggested the modification has been made.

Line 174. “worldwide [33].” Instead of “worldwide[33].”.

Reply: As suggested the modification has been made.

In section 2, there is no mention of chitin or its derivatives. These compounds are supposed to be the topic of the manuscript according to the title and the introduction. This mistake is repeated throughout the manuscript, so the title might be not appropriate. Otherwise, it is necessary to focus the information on this type of compounds.

Reply: In the review after introducing the cancers and the role of NLRP3inflammasome, chitin and its derivatives  have been  discussed in section 4.

Line 200 – 203. References missing

Reply: As suggested, the references have been added.

Line 204 – 205. Rewrite.

Reply: As suggested the sentence has been rewritten.

Line 207. “[41]. In” instead of “[41].In”.

Reply: As suggested the modification has been made.

Section 3.1. What does this section have to do with the dual role of chitin and its derivatives in nlrp3-mediated tumors? This is a mistake common to almost all the manuscripts. Table 1 has also the same problem.

Reply The section deals with the studies of other natural products in curtailing cancer. WE have included and retained it as we wanted to bring up the point of how studies have been initiated with other natural products and finally bring it to chitin.

Line 230 – 239. This paragraph is almost the only one that justifies the title of the manuscript in sections 2 and 3.

Reply:

Line 243. [55-57] instead of [55]–[57].

Reply: As suggested the modification has been made.

Line 254 – 255. Reference?

Reply: As suggested the modification has been made.

Line 261. “was”? Check verb tense.

Reply: As suggested the modification has been made.

Line 266-267. There cannot be paragraphs of only two lines.

Reply: As suggested the modification has been made.

Section 4.1. The information is disorganized and not well focused.

Reply: As suggested the corrections have been made in the section 4.1.

Line 339. “Utilized by us”. Rewrite. The same type of mistake in line 341 (“we can use…”).

Reply: As suggested the modification has been made

Line 345. “complications. In” instead of “complications). In”

Reply: As suggested the modification has been made.

Reviewer 4 Report

This manuscript describes a review on discussion regarding the role of Chitin and Its derivatives in NLRP3 Mediated Tumors. The studies, included in this context, suggest the possible usage of chitin and its derivatives as therapeutics to diseases that have inflammation driven or inflammation related complications. Therefore, I recommend publication of this manuscript after the following minor revisions.

1.       All abbreviations should be defined when they are first appeared in the text.

2.       At line 242, the authors mention that ‘Chitosan and its derivatives’. What derivatives are referred to?

3.       At line 281, a phase, ‘the deacetylated chitosan’, is not correct, because chitosan is already deacetylated.

4.       The effect of degrees of deacetylation (DDA) in chitosan on the immune response should be discussed, because DDA strongly affect the properties of chitosan.

5. English throughout the text should be checked and revised.

Author Response

Reply to Reviewer 4

  1. All abbreviations should be defined when they are first appeared in the text.

Reply: As suggested the modifications have been incorporated.

  1. At line 242, the authors mention that ‘Chitosan and its derivatives’. What derivatives are referred to?

Reply:  As suggested the modifications have been incorporated.

  1. At line 281, a phase, ‘the deacetylated chitosan’, is not correct, because chitosan is already deacetylated.

Reply:  As suggested the modifications have been incorporated.

  1. The effect of degrees of deacetylation (DDA) in chitosan on the immune response should be discussed, because DDA strongly affect the properties of chitosan.

Reply:  As suggested the role of DDA has been included and is highlighted.

  1. English throughout the text should be checked and revised.

Reply: As suggested the modifications have been incorporated.

Round 2

Reviewer 2 Report

Thank you for considering my comment on changing the title. In my opinion, it is now more suitable. 

Although you have made some changes in the text, including citing some more papers about chitin, my previous comment on "not existing a direct relationship between the text up to page 7 with chitin" is still there.

Author Response

The manuscript has tried to discuss the rule of chitin and its derivatives in NLRP3-mediated tumors. According to the text, it seems that it is NLRP3 itself that shows such behavior, while chitin and its derivatives have no direct effect on the tumors. Therefore I do not feel that the title of the manuscript has been chosen properly. 

Reply: In the gastrointestinal and gynaecological cancers we have included in the study, studies have reported that NLRP3 has a major role to play. What we have tried to emphasize   through  a collation of  different studies is that chitin is able to revert  the impact of NLRP3 and thereby can be used for  as a therapeutic effect. However not much study has been done on this aspect. We have tried to collate whatever studies have been initiated by various groups .  

2. By the way, most of the text deals with not related information to the title. For example up to page 7, one can not find any correlation between the rule of chitin and cancer. 

Reply: Initially we have tried to introduce different cancers and the role of NLRP3 in them and then finally in section 4 have tried to bring about the studies performed by various groups on the use of chitin in curtailing NLRP3 inflammasome,